# The effects of learning experience on college students' deep english learning: a study of the chain mediation effect of motivation and strategy

Qun Zhang[1,2¤a], Hao Jin [3¤b*], Wen Wen Teng[4¤c]

1 School of foreign language, An shun University, An shun, China, 2 School of Education, Arts, and Sciences, Lyceum of the Philippines University-Batangas, Philippines, 3 Jiangxi Tellhow Animation College, Nanchang, Jiangxi, China, 4 School of Innovation and Entrepreneurship, Guizhou Education University, Guiyang, China

¤a Current Address: Anshun City, Guizhou Province, China
¤b Current Address: Zhengzhou City, Henan Province, China
¤c Current Address: Guiyang City, Guizhou Province, China
* jinhao911216@163.com

## Abstract

This study focuses on the impact of learning experience on college students' deep learning of English and the chain-mediated effects of motivation and strategy. In the context of globalization, English is crucial for university students, but traditional teaching models often neglect the role of learning experience in deep learning. Deep learning emphasizes critical understanding, creative application and long-term memory construction, which is particularly important for English learning. Learning experience covers affective, cognitive and behavioral responses, and influences learning attitudes and effects, but there are fewer studies on its impact on college students' deep learning of English and the related mechanisms. In this study, college students of different genders, ages, educational backgrounds and academic achievement levels were selected as samples, and learning experience, motivation, learning strategies and deep learning were comprehensively assessed by well-designed scales and statistically analyzed with the help of SPSS and AMOS software. The results of the study show that learning experience has a significant positive effect on English deep learning, and motivation and learning strategies play an important chain mediating role. Specifically, learning experience enhances motivation, which in turn promotes the use of learning strategies and ultimately improves English deep learning. This study validates for the first time the chain mediation model of "learning experience→learning motivation→learning strategies→deep learning"in the field of English language learning, which provides a new perspective for understanding the intrinsic mechanism of college students' English language learning and enriches related research. In practice, it provides specific guidance for English teaching, and teachers can enhance students' English deep learning by optimizing learning experience, stimulating learning motivation and guiding the use of learning strategies. However, there

**Data availability statement:** All relevant data are included in the manuscript and its supporting information files.

**Funding:** The author(s) received no specific funding for this work.

**Competing interests:** The authors have declared that no competing interests exist.

are some limitations in this study, such as the limited sample scope and the use of a cross-sectional design, etc. Future studies can expand the sample scope, adopt a longitudinal research design, and further explore other potential mediating variables.

## 1. Introduction

In the context of globalization, the importance of English as an international common language is becoming more and more prominent. For university students, English is not only the basis for academic development, but also a key competence for professional development and personal growth [1]. However, the traditional English teaching model often focuses on the transmission of knowledge and neglects the role of learning experience in deep learning. Deep learning emphasizes critical understanding, creative application of knowledge and the construction of long-term memories [2], and this type of learning is particularly important in English language learning, as it can help students truly grasp the essence of the language and enhance their intercultural communication skills. In recent years, the concept of learning experience has gradually attracted the attention of educational researchers. Learning experience covers students' affective, cognitive and behavioral responses in the learning process, which not only affects students' attitudes towards learning, but also profoundly influences learning outcomes [3]. A positive learning experience can stimulate students' interest in learning, enhance learning motivation, and then promote the use of learning strategies and the improvement of learning outcomes. However, there is a relative paucity of research on how learning experience affects college students' deep learning of English. In particular, the role mechanisms of motivation and strategy in this process have not been fully explored [4].In educational psychology, motivation and strategy have always been key factors in the learning process. Motivation is the intrinsic motivation that drives students to learn, while learning strategies are the specific methods and techniques that students adopt to achieve their learning goals [5]. It has been shown that there is a significant positive correlation between motivation and strategies and learning outcomes [6]. However, the mediating role of motivation and strategies between learning experiences and learning outcomes, especially the chain mediation effect, has not been clearly elucidated. This chain mediation effect may reveal a complex learning process: the learning experience first affects students' motivation, which then influences the choice and use of learning strategies through the mediation of motivation, and ultimately affects learning outcomes. This chain mediation effect may be particularly important in the context of English language learning, which requires not only the accumulation of knowledge, but also the emotional engagement and flexible use of strategies [7].

This study aims to explore the impact of learning experience on college students' deep English learning and the chain mediating role of motivation and strategy in it through empirical research. To this end, college students with different genders, ages, educational backgrounds and academic achievement levels were selected as the study sample to ensure the broad applicability of the findings. Through well-designed scales, we conducted a comprehensive assessment of

learning experience, motivation, learning strategies and deep learning, and performed statistical analyses with the help of SPSS and AMOS software to ensure the scientific validity and reliability of the findings. In the course of the study, we focused on the following questions: (1) whether the learning experience has a direct effect on the deep learning of English; (2) whether motivation and strategy play the role of chain mediation between the learning experience and the deep learning; and (3) whether there are differences in the mediation pathways among different groups of students (e.g., with different levels of English proficiency). Compared with existing studies, this study is the first to validate the intermediary mechanism of the "motivation-strategy" chain in the field of English language learning and provides a theoretical basis for the design of the "context-motivation-strategy" trinity of instructional interventions. We expect that the findings of this study will deepen the understanding of the intrinsic mechanism of college students' English learning, provide valuable empirical support for English teaching practice, and offer new ideas and methods for optimizing teaching strategies.

## 2 Conceptual definition and research hypothesis

### 2.1 The relationship between learning experience and deep learning of English language

Learning experience is the result of college students' comprehensive perception of teaching methods, learning participation, learning evaluation and other teaching context factors in the process of English learning, covering both the teaching content and the teaching process, which is an organic combination of "what to learn" and "how to learn", emphasizing the active participation and personal experience of students in the learning process. It is an organic combination of "what to learn" and "how to learn", emphasizing students' active participation and personal experience in the learning process.In the context of English language teaching, rich English learning experiences are of great significance in changing students' learning styles and cultivating core academic qualities. Related studies show that learning experience is an important factor affecting students' deep learning, and positive learning experience can stimulate students' interest in learning, enhance their motivation, and then promote deep learning [8]. In addition, the theory of mind-flow points out that positive learning experiences (such as mind-flow experiences) can prompt students to spontaneously engage in learning activities without external motivation, and such experiences can effectively promote English learning by enhancing students' subjective sense of mastery and forming automaticity and spontaneity in learning [9].Therefore, this study proposes research hypothesis:

H1: English learning experience is significantly and positively related to English deep learning.

### 2.2 The Relationship between motivation, learning strategies and deeper learning of English

Learning motivation and learning strategies are key factors influencing students' deep learning. However, there are few empirical studies that directly explore the relationship between motivation, learning strategies and deep learning in English, and the existing studies focus more on the relationship between motivation, learning strategies and academic achievement. Academic achievement, as an important manifestation of the level of deep learning, can characterize the level of students' deep learning to a certain extent. It has been shown that both learning motivation and learning strategies are significantly and positively related to academic achievement. For example, Liu Jiaxia (2018) and Liu Zhihua (2019) found that both learning motivation and learning strategies of secondary school students are significantly positively correlated with academic achievement; Shen Xiajuan (2020) argued that students' learning motivation may have a direct or indirect effect on deep learning; Liu Mingjuan and Xiao Haiyan (2021) pointed out that students' motivation is significantly and positively correlated with academic achievement; Liu Dianzhi et al. pointed out that students' learning strategies significantly and positively predicted academic achievement, and that higher levels of learning strategies contribute to academic achievement. These studies provide a theoretical basis for exploring the roles of motivation and learning strategies in English deep learning [10].Therefore, this study proposes research hypothesis:

H2: English learning motivation and learning strategies are significantly and positively related to English deep learning.

## 2.3 The mediating role of motivation and learning strategies between learning experiences and deep learning of English

Based on social cognitive theory [11], students' learning experiences can have an impact on deep learning through the mediation of learning motivation and learning strategies. Specifically, learning experiences can enhance students' deep learning by stimulating motivation and facilitating the selection and use of learning strategies. It has been shown that contextual factors affect the acquisition of learning strategies, which in turn affects students' academic achievement; active classroom teaching can stimulate students' interest and motivation in learning, which in turn promotes deep learning [12]. In addition, motivation significantly influences students' choice and use of learning strategies [13].Therefore, this study proposes research hypothesis:

H3: Learning motivation and learning strategies play a chain mediating role between learning experience and deep English learning.

## 2.4 Research modelling

Based on the above three research hypotheses, this study constructed the following mediation model, as shown in Fig 1, in order to reveal the path of learning experience's influence on deep learning of English and its intrinsic mechanism:

The learning experience directly impacts deeper English language learning.

Motivation mediates between the learning experience and deep learning of English.

Learning strategies mediate between motivation and deep learning of English.

Learning experiences influence deep learning of English through the chain mediation of motivation and learning strategies.

This model will contribute to a deeper understanding of how learning experiences work through the mechanisms of motivation and learning strategies, ultimately contributing to students' deep learning of English.

## 3 Research methodology

### 3.1 Research sample

This study sourced its sample from both undergraduate and postgraduate students enrolled in various comprehensive universities located in Nanchang City, Jiangxi Province, China. The sampling specifically focused on individuals with educational backgrounds spanning from the bachelor's level up to and including doctoral studies.The purpose was to scrutinize the interconnections between English learning experiences, motivational factors, learning strategies, and deep learning outcomes within the higher education framework.To guarantee the broad applicability of the research findings, the sample was meticulously constructed to incorporate students of diverse genders, ages, and academic accomplishments. More precisely, the sample comprised male and female students aged 18 and older, with educational qualifications ranging from bachelor's degrees through to doctoral degrees. The gender distribution was balanced, nearing a 1:1 ratio, and the sample featured a wide age range as well as varied academic achievements across different year levels.In pursuit of ensuring the representativeness of the sample, a stratified sampling technique was implemented. Students were categorized

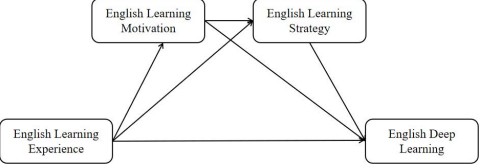

**Fig1. Research hypothesis model.**

according to their year of study, field of specialization, and English language proficiency. This methodological approach served to diminish sampling errors and augment the dependability of the research outcomes. A total of 500 questionnaires were disseminated, with 450 valid responses retrieved, translating to an effective response rate of 90%. This diverse and substantial sample cohort offers a robust foundation for the subsequent statistical analyses.. Specifically as shown in Table 1.

In this study, a questionnaire was administered to the participants to collect data on the relationship between learning experiences, motivation and strategies and deep learning outcomes. The period of the study was from 1st October 2024–31st January 2025. All participants were informed of the purpose and content of the study prior to its commencement and they all voluntarily consented to participate. Participants were all adults and therefore did not need to obtain consent from their parents or guardians. The study qualified for an exemption from ethical review because it collected data through an anonymous questionnaire, did not involve interventions or manipulations that could negatively affect the physical or mental health of the participants, and the data did not contain any personally identifiable information, nor did it involve sensitive personal information or commercial interests. According to Article 32, Chapter 3 of the Measures for Ethical Review of Life Science and Medical Research Involving Human Beings by the Central People's Government of the People's Republic of China, this study met the conditions for exemption from ethical review. The study collected data through anonymous questionnaires, did not involve interventions or manipulations that might negatively affect the physical and mental health of the participants, and the data did not contain any personally identifiable information, nor did it involve sensitive personal information or commercial interests.

### 3.2 Scale design

In order to accurately measure the research variables, this study designed four scales to assess English learning experience, motivation, learning strategies and deep learning outcomes. The scale design is based on existing research and theoretical frameworks to ensure that each scale has clear dimensions, reasonable items, and good reliability [14]. The scales are rated on a five-point scale (1 = strongly disagree, 5 = strongly agree), with higher scores indicating more positive corresponding experiences, motivations, strategies, or learning outcomes. The specific design of each scale is presented below:

**3.2.1 English learning experience scale.** The English Learning Experience Scale (ELES) is designed to comprehensively assess students' subjective experience in the process of English learning, covering three dimensions [15]. The Teaching Style dimension focuses on the effectiveness and interactivity of teaching methods, such as "The teacher's teaching methods help me to understand English"; the Learning Engagement dimension measures how active students are in the classroom, such as "I actively participate in class discussions"; and the Learning Evaluation dimension collects students'

**Table 1. Descriptive statistical analysis of the sample.**

| variant | form | Frequency (n) | Percentage (%) |
|---|---|---|---|
| distinguishing between the sexes | a male | 225 | 50 |
| | females | 225 | 50 |
| (a person's) age | 1820 years | 120 | 26.7 |
| | 2125 years old | 180 | 40 |
| | 2630 years old. | 90 | 20 |
| | 31 and over | 60 | 13.3 |
| educational background | Technical Student | 30 | 6.7 |
| | undergraduate (adjective) | 240 | 53.3 |
| | bachelor's degree | 120 | 26.7 |
| | PhD and above | 60 | 13.3 |

feedback on the course, such as "I actively participate in class discussions". The learning participation dimension measures how active students are in class, e.g., "I actively participate in class discussions"; and the learning evaluation dimension collects students' feedback on the course, e.g., "I am satisfied with this semester's English course". The scale consists of 15 items, and the item design of each dimension is based on the existing research [16], and the reliability and validity have been verified by pretest. The Cronbach's alpha coefficient of the scale was 0.87, indicating good internal consistency.

### 3.2.2 English learning motivation scale.

The English Learning Motivation Scale measures intrinsic and extrinsic factors that drive students to learn English and is based on Dörnyei's (2020) theory of motivation. The scale is divided into two dimensions: intrinsic motivation, which is related to personal interest and self-improvement, e.g., "I am interested in learning English"; and extrinsic motivation, which is related to external rewards or pressures, e.g., "I am learning English in order to pass the exam". The scale consists of 10 items and is rated on a five-point scale. The Cronbach's alpha coefficient of the scale was 0.85, indicating high reliability. In addition, the structural validity of the scale was verified by exploratory factor analysis (EFA) to ensure that it accurately reflects the two dimensions of motivation to learn [17].

### 3.2.3 English learning strategies scale.

The English Learning Strategies Scale (ELSS) assesses the different strategies that students use in the process of learning English, covering three dimensions: cognitive strategies, metacognitive strategies and resource management strategies (Oxford, 2016). Cognitive strategies focus on information processing and memorization skills, e.g. "I will memorize English words through repetition"; metacognitive strategies involve learning planning and self-monitoring, e.g. "I will regularly assess my learning progress"; and resource management strategies focus on the use of additional resources to support learning, e.g. "I will use the English language to learn English. Metacognitive strategies involve learning planning and self-monitoring, e.g., "I will regularly assess my learning progress"; resource management strategies focus on using additional resources to support learning, e.g., "I will use online learning platforms to improve my English language skills". The scale consists of 20 items, and the Cronbach's alpha coefficient is 0.89, indicating good reliability. The structural validity of the scale was further validated by a validated factor analysis (CFA) to ensure that it could accurately measure the three dimensions of learning strategies [18].

### 3.2.4 English deep learning scale.

The Deeper Learning in English Scale (DLES) is designed to measure students' deeper learning in the English language subject and covers the cognitive, interpersonal and personal domains. The cognitive domain covers knowledge construction and problem solving, for example, "I can apply my English knowledge to real-life problems"; the interpersonal domain covers collaboration and communication, for example, "I can communicate effectively with others in English"; and the personal domain focuses on independent learning and perseverance, for example, "I can complete English learning tasks independently". The personal domain focuses on independent learning and perseverance, such as "I am able to complete English learning tasks independently". The scale consists of 12 items, and the Cronbach's alpha coefficient is 0.86, indicating high reliability. The structural validity of the scale was validated by EFA and CFA to ensure that it accurately reflects the three domains of deep learning [19].

## 3.3 Statistical processing

SPSS 27.0 and AMOS 27.0 software were used to analyze the data in this study to ensure the scientificity and reliability of the statistical analysis. Firstly, descriptive statistical analysis was used to provide a preliminary understanding of the distribution of the variables, including mean, standard deviation and frequency distribution. Secondly, correlation analysis was used to test the correlation

between the variables to provide a basis for the initial validation of the hypotheses (Field, 2018). Subsequently, structural equation modelling (SEM) analysis was used to verify the relationship between learning experience, motivation, learning strategies and deep English learning, and to test the chain mediation effect of motivation and learning strategies [20]. The model fit indicators include $\chi^2/df$, RMSEA, CFI and TLI to ensure the reasonableness and fit of the mode. Through this series of analyses, this study was able to systematically reveal the relationship between the variables and their mechanisms of action in English deep learning.

# 4 Findings

## 4.1 model checking

**4.1.1 Reliability and validity tests.** In this study, the reliability and validity of all scales were thoroughly tested to ensure the scientific validity and reliability of the measurement instruments. The reliability test was assessed by Cronbach's alpha coefficient, factor loadings and Composite Reliability (CR), while the validity test included both structural and discriminant validity.

**4.1.2 Reliability test.** Reliability is an important indicator of scale reliability, which was assessed in this study using Cronbach's α coefficient, factor loadings and combined reliability (CR). The Cronbach's α coefficients of all the scales are higher than 0.7, indicating that the scales have good internal consistency [27]. Specifically, the Cronbach's α coefficients for English learning experience (α = 0.907), English learning motivation (α = 0.837), English learning strategies (α = 0.890), and English deep learning (α = 0.907) were at a high level. In addition, the factor loadings of each scale were higher than 0.6, further validating the reliability of the scales. The combined reliability (CR) values were all above 0.7, further supporting the reliability of the scales [28]. These results indicate that the scales in this study have high reliability in measuring the corresponding constructs.

**4.1.3 Validity tests.** Validity tests include both structural validity and discriminant validity. Structural validity was assessed by standardised factor loadings, combined reliability (CR) and average extracted variance (AVE). The standardised factor loadings of each scale were higher than 0.6, the CR values were higher than 0.7, and the AVE values were higher than 0.5, indicating that the scales have good convergent validity [29]. The specific data are shown in Table 2:

The discriminant validity was tested by the method proposed by Fornell and Larcker. The results of the calculation showed that the square root of the AVE in each column was greater than the correlation coefficient between the variables, indicating that the scale had good discriminant validity (as shown in Table 3). For example, the square root of AVE = 0.551 for English learning experience is 0.742, which is higher than the correlation coefficients with other variables (e.g., the correlation coefficient with motivation to learn English is 0.209).

## 4.2 Common methodology bias test

In order to minimise the possibility of common method bias, the purpose of the study was clearly stated to the participants in the questionnaire and the anonymity of the responses was emphasised to reduce their concerns. Finally, the collected data were examined using the Harman one-way test to rule out the potential effect of common method bias. The results of the test showed that the eigenvalues of the four factors involved were all greater than 1 and the maximum factor variance explained was 21.012%, which is below the 40% threshold [21] This indicates that there is no serious common method bias in the data of this study.

## 4.3 Model fit tests

Validated factor analysis (CFA) was conducted using AMOS 24.0 to test the fit of the four-factor model (ELL experience, ELL motivation, ELL strategies, and ELL deep learning). The four-factor model fit the data better than the other competing models. The model fit indicators are shown in Table 4:

All the indicators meet the measurement criteria, indicating that the overall fit of the four-factor model is good [22]. The $\chi^2$/DF value is 1.666 (less than 3), the RMSEA value is 0.04 (less than 0.08), and the values of GFI, CFI, NFI, and IFI are all higher than 0.9, which further supports the reasonableness of the four-factor model. Specifically as shown in Fig 2.

## 4.4 Direct effects tests

Based on the good model fit, AMOS 24.0 was used to conduct the model path test. The results show that English learning experience has a significant positive effect on English learning motivation (Estimate = 0.211, $p < 0.001$) [23],

**Table 2. Results of reliability and validity analyses of the scale.**

| Variable | Item | Factor Loading | CR | AVE | Alpha |
|---|---|---|---|---|---|
| English Learning Experience | 1. I think the teaching method in English classrooms is very effective. | 0.746 | 0.907 | 0.551 | 0.907 |
| | 2. The teaching methods of English teachers can stimulate my interest. | 0.790 | | | |
| | 3. English classroom activities make me feel interesting. | 0.689 | | | |
| | 4. I actively participate in discussions in English class. | 0.724 | | | |
| | 5. I think English homework is helpful for my learning. | 0.724 | | | |
| | 6. I often have the opportunity to express my opinions in English class. | 0.770 | | | |
| | 7. The feedback from the English exam helped me understand my learning progress. | 0.759 | | | |
| | 8. I am able to understand my shortcomings through the evaluation of my English class learning. | 0.730 | | | |
| English Learning Motivation | 9. I have an inherent interest in learning English. | 0.726 | 0.838 | 0.509 | 0.837 |
| | 10. Learning English is a pleasure for me. | 0.647 | | | |
| | 11. Learning English can help me obtain better job opportunities. | 0.717 | | | |
| | 12. By learning English, I can better understand the world. | 0.732 | | | |
| | 13. I study English to improve my social skills. | 0.741 | | | |
| English Learning Strategy | 14. I memorise new words by making vocabulary cards. | 0.679 | 0.890 | 0.504 | 0.890 |
| | 15. I often use an English dictionary to help understand unfamiliar words. | 0.739 | | | |
| | 16. I improve my reading comprehension ability by summarising the main idea of paragraphs. | 0.743 | | | |
| | 17. I set clear learning goals when learning English. | 0.720 | | | |
| | 18. I check my learning effectiveness through self testing. | 0.732 | | | |
| | 19. I use resources such as English corners to improve my speaking ability. | 0.707 | | | |
| | 20. I improve my listening skills by watching English movies. | 0.657 | | | |
| | 21. I improve my English proficiency by collaborating with others for learning. | 0.697 | | | |
| English Deep Learning | 22. Even if I encounter difficulties, I can persist in learning English. | 0.783 | 0.907 | 0.620 | 0.907 |
| | 23. I am able to independently complete English learning tasks. | 0.791 | | | |
| | 24. I am able to express my ideas clearly in English. | 0.747 | | | |
| | 25. I am able to collaborate with others to complete English learning tasks. | 0.811 | | | |
| | 26. I am able to write creatively in English. | 0.814 | | | |
| | 27. I engage in critical thinking when learning English. | 0.777 | | | |

supporting Hypothesis 1; English learning motivation has a significant positive effect on English learning strategies (Estimate = 0.271, p < 0.001), supporting Hypothesis 2; English learning strategies have a significant positive influence (Estimate = 0.357, p < 0.001), supporting Hypothesis 3; and English learning experience has a significant positive influence (Estimate = 0.314, p < 0.001) on English deep learning, supporting Hypothesis 4. The specific results are shown in Table 5:

**Table 3. Results of the discriminant validity test.**

| Variable | English Learning Experience | English Learning Motivation | Learning Strategy | English Deep Learning |
|---|---|---|---|---|
| English Learning Experience | *0.742* | | | |
| English Learning Motivation | 0.209** | *0.713* | | |
| Learning Strategy | 0.373** | 0.367** | *0.710* | |
| English Deep Learning | 0.407** | 0.328** | 0.439** | *0.787* |
| AVE | 0.551 | 0.509 | 0.504 | 0.620 |

Note: * indicates P < 0.01, ** indicates P < 0.01.

**Table 4. Indicators of model fit.**

| MODEL | $\chi^2$ | DF | $\chi^2$/DF | GFI | NFI | IFI | CFI | RMSEA |
|---|---|---|---|---|---|---|---|---|
| Four-Factor Model (ELE, ELM, ELS, EDL) | 529.759 | 318 | 1.666 | 0.901 | 0.911 | 0.963 | 0.958 | 0.04 |
| Three Factors Model (ELE, ELM and ELS, EDL) | 1125.743 | 321 | 3.507 | 0.79 | 0.812 | 0.858 | 0.857 | 0.078 |
| Two Factors Model (ELE and ELM and ELS, EDL) | 2333.958 | 323 | 7.226 | 0.552 | 0.609 | 0.644 | 0.642 | 0.123 |
| Single-Factor Model (ELE and ELM and ELS and EDL) | 3231.187 | 324 | 9.973 | 0.475 | 0.459 | 0.485 | 0.483 | 0.147 |

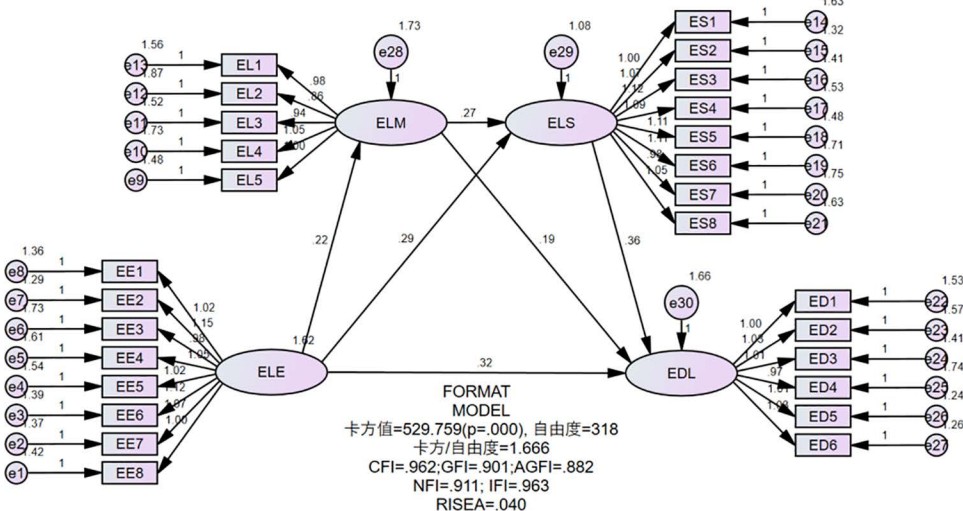

**Fig 2. Model results.**

**Table 5. Direct effect test results.**

| Route | Estimate | SE | P |
|---|---|---|---|
| English Learning Experience→ English Learning Motivation | 0.211 | 0.057 | *** |
| English Learning Motivation→ English Learning Strategy | 0.271 | 0.051 | *** |
| English Learning Experience→ English Learning Strategy | 0.280 | 0.050 | *** |
| English Learning Strategy→ English Deep Learning | 0.357 | 0.076 | *** |
| English Learning Experience→ English Deep Learning | 0.314 | 0.063 | *** |
| English Learning Motivation→ English Deep Learning | 0.198 | 0.064 | 0.002 |

Note: * indicates P < 0.01, ** indicates P < 0.01.

## 4.5 Testing for intermediation effects

Bootstrap method was used to test the mediating effect, AMOS software provides the corresponding calculation function, the sample size of Bootstrap was chosen to be 5000, and the confidence interval was 95% [24]. The test results show that the indirect effect of English learning experience on English deep learning through English learning motivation is significant (Estimate = 0.409, p = 0.000); the indirect effect of English learning experience on English deep learning through English learning strategies is significant (Estimate = 0.637, p = 0.000); the indirect effect of English learning experience on English deep learning through English learning motivation and English learning strategies has a significant chain-mediated effect on English deep learning (Estimate = 0.839, p = 0.000). The specific results are shown in Table 6:

## 5 Discussion

This study explored the influence of learning experience on college students' English deep learning through empirical analyses, and the chain-mediated roles of learning motivation and learning strategies in the process. The findings indicate that learning experience has a significant positive effect on English deep learning, and that learning motivation and learning strategies play an important chain mediating role in this process. These findings not only validate the hypotheses of this study, but also provide valuable theoretical support for English teaching practice.

### 1. The direct impact of the learning experience on deeper English language learning

The findings show that learning experience has a significant positive effect on deep learning of English (Estimate = 0.314, p < 0.001), which suggests that a positive learning experience can directly contribute to students' deep learning of English. This finding is consistent with [25] states that learning experiences can significantly increase students' motivation and engagement, which in turn promotes deep learning. Therefore, this study emphasises the importance of learning experience in English language teaching and suggests that teachers should enhance students' learning experience through diversified teaching methods and interactive classroom design, thereby promoting deep learning [26].

### 2. Chain mediation of motivation and learning strategies

The present study further reveals the chain mediating role of motivation and learning strategies between learning experience and deep learning of English. Specifically, learning experience enhanced deep English learning (Estimate = 0.357, p < 0.001) by enhancing motivation (Estimate = 0.211, p < 0.001) [27], which in turn facilitated the use of learning strategies (Estimate = 0.271, p < 0.001), and ultimately, deep English learning (Estimate = 0.357, p < 0.001). The significance of this chain-mediated effect (Estimate = 0.839, p = 0.000) further validates the role of motivation and learning strategies in bridging the gap between learning experiences and deep learning [28].

This finding is consistent with's theory of motivation, which states that motivation is the key factor that drives students to actively participate in learning activities. Meanwhile, pointed out that effective learning strategies can significantly enhance learning outcomes. Therefore, this study suggests that in English language teaching [29] teachers should not

**Table 6. Results of the mediation effects test.**

| Route | Estimate | Lower | Upper | P |
|---|---|---|---|---|
| English Learning Experience→ English Learning Motivation→ English Deep Learning | 0.409 | 0.250 | 0.577 | 0.000 |
| English Learning Experience→ English Learning Strategy→ English Deep Learning | 0.637 | 0.458 | 0.819 | 0.000 |
| English Learning Experience→ English Learning Motivation→ English Learning Strategy→ English Deep Learning | 0.839 | 0.636 | 1.053 | 0.000 |

Note: * indicates P < 0.01, ** indicates P < 0.01.

only pay attention to students' learning experiences, but also promote students' deep learning by stimulating motivation and guiding the use of learning strategies.

### 3. Theoretical and practical implications of the study

The theoretical significance of this study lies in the validation of the chain mediation model of "learning experience → learning motivation → learning strategy → deep learning" for the first time in the field of English learning. This model provides a new perspective for understanding the intrinsic mechanism of college students' English learning and enriches the research on the relationship between learning experience and deep learning. In addition, the practical significance of this study lies in the fact that it provides specific guidance for English teaching. By optimising the learning experience, stimulating motivation, and instructing students to use learning strategies effectively, teachers can significantly improve students' English deep learning.

### 4. Limitations of the study and directions for future research

Despite the meaningful findings of this study, there are some limitations. First, the sample was limited to undergraduates of a comprehensive university, which may not be fully representative of all college student populations. Future research could expand the sample to cover different regions and types of colleges and universities to enhance the generalisability of the findings. Second, this study used a cross-sectional design, which could not completely exclude the reverse effect of causality. Future studies could adopt a longitudinal research design to track the process of changes in students' learning experiences, motivation, strategies and deep learning. In addition, future studies could further explore other potential mediating variables, such as learning anxiety and learning burnout, to more fully understand the complex relationship between learning experience and deep learning.

## 6 Conclusion

This study reveals through empirical analyses the significant positive influence of learning experience on college students' deep learning of English, as well as the chain mediating roles of motivation and learning strategies in it. These findings not only provide valuable guidance for English teaching practice, but also offer new directions for future research. By optimising the learning experience, stimulating motivation and guiding the use of learning strategies, teachers can significantly enhance students' deep English learning, thus laying a solid foundation for their academic development and professional growth.

## Supporting information

**S1 Data. Primary information collected directly during the course of the study.**
(XLSX)

## Acknowledgments

We would like to thank all participants for their contributions to this study.

## Author contributions

**Conceptualization:** Hao Jin.

**Data curation:** Hao Jin.

**Formal analysis:** Hao Jin.

**Methodology:** Qun Zhang.

**Project administration:** Qun Zhang.

**Software:** Wen Wen Teng.

**Validation:** Wen Wen Teng.

**Writing – original draft:** Qun Zhang.

**Writing – review & editing:** Wen Wen Teng.

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
