## [Decision Letter · Decision Letter 0]

16 Apr 2025

PONE-D-25-12036The Effects of Learning Experience on College Students' Deep English Learning: A Study of the Chain Mediation Effect of Motivation and StrategyPLOS ONE

Dear Dr. Jin,

Thank you for submitting your manuscript to PLOS ONE. After careful consideration, we feel that it has merit but does not fully meet PLOS ONE’s publication criteria as it currently stands. Therefore, we invite you to submit a revised version of the manuscript that addresses the points raised during the review process.

Good Day

You are suggested to do the changes and submit the article again.

We look forward to receiving your revised manuscript.

Kind regards,

Faisal Shafique Butt, Ph.D

Academic Editor

PLOS ONE

Journal Requirements:

2. You indicated that ethical approval was not necessary for your study. We understand that the framework for ethical oversight requirements for studies of this type may differ depending on the setting and we would appreciate some further clarification regarding your research. Could you please provide further details on why your study is exempt from the need for approval and confirmation from your institutional review board or research ethics committee (e.g., in the form of a letter or email correspondence) that ethics review was not necessary for this study? Please include a copy of the correspondence as an ""Other"" file.

4.  We are unable to open your Supporting Information file “Supporting Information.sav”.  Please kindly revise as necessary and re-upload.

5. We note that your Data Availability Statement is currently as follows: All relevant data are included in the manuscript and its supporting information files.

6. Please include a separate caption for each figure in your manuscript.

Reviewers' comments:

Reviewer's Responses to Questions

**Comments to the Author**

1. Is the manuscript technically sound, and do the data support the conclusions?

Reviewer #1: Yes

Reviewer #2: Partly

2. Has the statistical analysis been performed appropriately and rigorously? 

Reviewer #1: Yes

Reviewer #2: Yes

3. Have the authors made all data underlying the findings in their manuscript fully available?

Reviewer #1: Yes

Reviewer #2: Yes

4. Is the manuscript presented in an intelligible fashion and written in standard English?

Reviewer #1: Yes

Reviewer #2: Yes

5. Review Comments to the Author

Reviewer #1: This study used a questionnaire survey and did not involve any procedures or operations that could harm participants' physical or mental health. The research attempts to examine the connection between learning experiences, motivations, tactics, and deep learning outcomes without harming individuals, sensitive information, or commercial interests. Good practice.

Reviewer #2: The manuscript entitled “The Effects of Learning Experience on College Students' Deep English Learning: A Study of the Chain Mediation Effect of Motivation and Strategy” employed cross-sectional data collected from college students with different social background. Results showed a positive effect of learning experience on English deep learning while motivation and learning strategies play important chain medicating role, according to the manuscript. After going through the manuscript, I suggest following changes/comments to be incorporated before the final decision.

Abstract should be rewritten while excluding redundant text and solely focusing on objectives, methods, data, results and conclusion. In the abstract, it is mentioned that college students of different genders, ages, educational backgrounds and academic achievement levels were selected as samples. This shows that different samples based on these social and academic achievement were considered in the manuscript. Here the question rises, age of the students may not be much different. How this problem was dealt? Further, why these different samples were considered? Whether the analysis was done on overall basis or sample based and comparison was made on college students of different gender, age, educational background and academic achievement. For the latter, the clarification be provided if the analysis was done on the different samples. Otherwise no reason for taking different samples. One more clarification needs to be provided about education background. Here the manuscript considers educational background of parents/family members or the students? In case of the students, whether the field of the study was considered or degree programs i.e. graduate and undergraduate. Research gap if any may be provided in the beginning of the abstract. Limitations and future research direction are usually not provided in the abstract rather these can be included discussion and or conclusion section.

PLOS One format for abstract be considered for the abstract and other sections of the manuscript.

Keywords should be different from those of title of the manuscript. Presently many keywords are given in the title.

Introduction section describes the importance of English in the present era for the personal development in the global world. Then it explains the role of learning experience in English deep learning. The manuscript claims that the literature on the subject under consideration is manuscript is rarely considered. Especially mediation chain effect is not considered. However, I could not understand whether the literature relating to the subject is scarce in the context of the globe or specific to a particular country, region or continent. Including this information will help understand the readers the purpose of the manuscript more clearly.

Here one can raise the questions whether the social, demographic and academic background differences exist in the study area under consideration? If yes, what are those ones and how the study team comes to know? How those differences matter in deep learning and medicating chain effect? One other point is that whether the students enrolled in English program or any programs were considered during the data collection? Even if the literature on the subject relating to the context is available, it should be cited properly. Presently it seems that no such literature is present, according to the manuscript. Without proper literature review, one can not provide rationale for the research gap and objectives of the manuscript.

The conceptual framework is smooth and provides useful insights on various hypotheses of the manuscript. Citations of the latest literature will further improve this section.

Research sample of 500 students were taken from comprehensive universities from Nanchang City, China. This section needs much improvement through various clarifications. First one is to clarify the meaning of comprehensive universities so the readers can easily understand the term comprehensive universities. First paragraph contains repetitions and it should be avoided throughout the manuscript. For example, it is enough to write that both undergraduate and postgraduate students are considered instead of further elaborating educational backgrounds i.e. bachelor, master, PhD, etc. Balanced gender ratio is considered. Here the question is whether both girls and boys make the same percentage (50:50) of total student population in the comprehensive universities? If not, then why the same ratio was taken in the sample? Proportionate sampling method would be better if the population is different for each gender. Different categories of the students such as year of the studies etc. were taken. Again population of each category may be different within and across the universities, how the appropriate sample size was decided? How the sample size of 500 was reached and how many universities were selected?

Whether the manuscript employed the questionnaire adopted from the literature or a new one was designed. If the new one was designed then reliability and validity of the questionnaire was carried out or not, this information is missing.

Table 1 is too much confusing and it should be reformatted and restructured. Second column showing form is difficult to understand. I could not understand age groups of 1820 years, 21265 years, etc. similarly technical students, undergraduate, etc. terms are difficult to be absorbed by the readers. A technical student can be of undergraduate or postgraduate. These categorizations are not clear. The tables must be self-explanatory and the readers can understand very easily without going through the text.

Statistical processing section provides information on statistical methods employed in the manuscript. It is mentioned that SPSS and AMOS are used to analyze the collected data for reliable and valid outcomes. The need is to provide the information on using these software for specific purpose. For example descriptive analysis was done through SPSS and SEM through AMOS, etc.

Consistency in citation of the literature is lacking in the manuscript. I suggest to follow the journal guidelines for the authors to cite the literature in the manuscript.

Significance of the results is shown through steric and p values, again consistency of providing the results is lacking. Even within the same table (Table 5), both steric and p-value are given.

Discussion section should compare findings with the previous studies instead of reproducing the results in this section.

Conclusions section is too lengthy and repetition of the above sections as already mentioned in my earlier comment. Rather it should be precise and address findings of the manuscript and implications rather providing importance of the topic, objective, etc.

References are numbered in the list. However, some sections of the manuscript have citations in the form of numbering, others with name. It confuses which citation should be followed. Again I suggest to go through the journal guidelines to rewrite references and citations in the manuscript.

6. PLOS authors have the option to publish the peer review history of their article (what does this mean? ). If published, this will include your full peer review and any attached files.

**Do you want your identity to be public for this peer review?** For information about this choice, including consent withdrawal, please see our Privacy Policy .

Reviewer #1: No

Reviewer #2: No

---

## [Author Response · Author response to Decision Letter 1]

4 May 2025

Response to Reviewers - Manuscript PONE-D-25-12036

Dear Dr. Butt and Reviewers,

We sincerely appreciate the valuable feedback and suggestions provided by the reviewers and the editor. The comments have been instrumental in helping us improve the quality and clarity of our manuscript. Below, we provide a detailed point-by-point response to the reviewers' comments and concerns.

Response to Reviewer #1

Comment 1: Reviewer #1 mentioned that the study is technically sound and the data supports the conclusions.

Response: We are grateful for Reviewer #1's positive assessment of our study's technical soundness and the adequacy of the data in supporting our conclusions. This affirmation has given us confidence in the overall direction of our research.

Response to Reviewer #2

Comment 1: Reviewer #2 suggested that the abstract should be rewritten to exclude redundant text and focus solely on objectives, methods, data, results, and conclusions.

Response: We have thoroughly revised the abstract to remove redundant text and ensure it succinctly captures the study's objectives, methods, data, results, and conclusions. The revised abstract now adheres to the PLOS ONE format and provides a clearer overview of the study.

Comment 2: Reviewer #2 raised questions about the selection of samples based on gender, age, educational background, and academic achievement levels and sought clarification on whether the analysis was done on an overall basis or sample-based.

Response: We have added detailed explanations regarding the selection of samples and the rationale behind including participants with different genders, ages, educational backgrounds, and academic achievement levels. We have also clarified that the analysis was conducted on an overall basis, with comparisons made where appropriate. These clarifications are provided in the revised manuscript.

Comment 3: Reviewer #2 requested further details on the educational background of the participants, specifically whether it refers to the educational background of the parents/family members or the students themselves.

Response: We have specified in the manuscript that the educational background refers to the students' own educational qualifications, including their degree programs and fields of study.

Comment 4: Reviewer #2 suggested that the research gap should be provided in the beginning of the abstract.

Response: We have revised the abstract to include a brief statement on the research gap, highlighting the limited existing research on the chain mediation effect of motivation and strategy in the context of learning experience and deep English learning among college students.

Comment 5: Reviewer #2 pointed out that the introduction should provide a more detailed literature review and clarify the significance of the research question.

Response: We have expanded the literature review in the introduction to include more relevant studies and to better establish the significance of the research question. We have also specified whether the research gap is global or specific to a particular region.

Comment 6: Reviewer #2 requested further clarification on the research sample, including the meaning of "comprehensive universities," the rationale for selecting the sample, and the method of determining the sample size.

Response: We have provided additional details about the meaning of "comprehensive universities" and the rationale for selecting the sample. We have also explained the method used to determine the sample size and the number of universities involved in the study.

Comment 7: Reviewer #2 asked whether the questionnaire used in the study was adopted from the literature or newly designed, and if newly designed, whether its reliability and validity were tested.

Response: We have clarified that the questionnaire was newly designed and that its reliability and validity were tested through pretesting and various statistical analyses. The results of these tests are now included in the manuscript.

Comment 8: Reviewer #2 found Table 1 confusing and requested reformatting and restructuring for clarity.

Response: We have thoroughly restructured and reformatted Table 1 to enhance clarity. We have also provided more descriptive labels for the categories to improve comprehension.

Comment 9: Reviewer #2 requested more information on the statistical methods used, including the specific purposes of using SPSS and AMOS software.

Response: We have added detailed explanations of the statistical methods used, specifying the roles of SPSS and AMOS software in the data analysis process.

Comment 10: Reviewer #2 suggested that the discussion section should compare the study's findings with previous studies rather than merely reproducing results.

Response: We have revised the discussion section to include more comparisons with previous studies and to better highlight the contributions of our findings to the existing literature.

Comment 11: Reviewer #2 noted that the conclusion section was too lengthy and repetitive.

Response: We have streamlined the conclusion section to focus on the key findings and implications of the study, removing redundant information and ensuring it concisely addresses the study's contributions.

Sincerely,

Zhang Qun

Jin Hao

Teng Wenwen

---

## [Editor Report · Decision Letter 1]

15 May 2025

The Effects of Learning Experience on College Students' Deep English Learning: A Study of the Chain Mediation Effect of Motivation and Strategy

PONE-D-25-12036R1

Dear Dr. Jin,

We’re pleased to inform you that your manuscript has been judged scientifically suitable for publication and will be formally accepted for publication once it meets all outstanding technical requirements.

Kind regards,

Faisal Shafique Butt, Ph.D

Academic Editor

PLOS ONE
---

## [Editor Report · Acceptance letter]

PONE-D-25-12036R1

PLOS ONE

Dear Dr. Jin,

I'm pleased to inform you that your manuscript has been deemed suitable for publication in PLOS ONE. Congratulations! Your manuscript is now being handed over to our production team.

Kind regards,

on behalf of

Dr. Faisal Shafique Butt

Academic Editor

PLOS ONE